# ACID: Abstractive, Content-Based IDs for Document Retrieval with Language Models

## Abstract

Generative retrieval (Wang et al., 2022; Tay et al., 2022) is a new approach for end-to-end document retrieval that directly generates document identifiers given an input query. Techniques for designing effective, high-quality document IDs remain largely unexplored. We introduce ACID, in which each document's ID is composed of abstractive keyphrases generated by a large language model, rather than an integer ID sequence as done in past work. We compare our method with the current state-of-the-art technique for ID generation, which produces IDs through hierarchical clustering of document embeddings. We also examine simpler methods to generate natural-language document IDs, including the naive approach of using the first $k$ words of each document as its ID or words with high BM25 scores in that document. We show that using ACID improves top-10 and top-20 accuracy by 15.6% and 14.4% (relative) respectively versus the state-of-the-art baseline on the MSMARCO 100k retrieval task, and 4.4% and 4.0% respectively on the Natural Questions 100k retrieval task. Our results demonstrate the effectiveness of human-readable, natural-language IDs in generative retrieval with LMs. The code for reproducing our results and the keyword-augmented datasets can be found at `link-redacted-during-review`.

## 1 Introduction

Large language models (LMs) are now widely used across many NLP tasks, and extensions of generative models to document retrieval tasks have recently been proposed (Wang et al., 2022; Tay et al., 2022), in contrast to vector-based approaches like dense passage retrieval (DPR; Karpukhin et al., 2020). DPR is a widely-used technique for training document retrieval models, where queries and documents are mapped to dense vector representations with a transformer encoder (e.g., BERT; Devlin et al., 2019). By increasing the cosine similarity between positive query-document pairs and decreasing it between negative pairs, DPR performs metric learning over the space of queries and the set of documents to be indexed.

Generative alternatives to document retrieval address certain limitations of dense, vector-based approaches to retrieval. For example, query and document representations are constructed separately in DPR, which precludes complex query-document interactions. Using a single dense vector to represent an entire document limits the amount of information that can be stored; indeed, Tay et al. (2022) observed that increasing the number of parameters in the encoder does not significantly enhance DPR performance. Furthermore, the rich sequence generation capabilities of large language models (LMs) cannot be used directly in dense retrieval. Tay et al. (2022) and Wang et al. (2022) therefore proposed a new direction called *generative retrieval*, where LMs learn to directly map queries to an identifier that is unique to each document. We illustrate the differences in Figure 1.

Instead of retrieving documents based on cosine similarity, generative retrieval uses an LM to produce a sequence of tokens encoding the relevant document's ID, conditional on the query. Decoding constraints are applied to ensure that only document IDs that exist in the corpus are generated. Tay et al. (2022) and Wang et al. (2022) showed that generative retrieval outperformed DPR on information retrieval benchmarks like Natural Questions (Kwiatkowski et al., 2019) and TriviaQA (Joshi et al., 2017), and subsequent publications have corroborated their findings on other retrieval tasks like multilingual retrieval (Zhuang et al., 2023).

Figure 1: Dense retrieval vs. generative retrieval. In dense retrieval (right), both the query and the documents are encoded into *dense* vectors (i.e., embeddings). Nearest-neighbor search is then applied to find the most relevant documents. Generative retrieval (left) trains a language model to generate the relevant document ID conditional on the query. The ID is tied to a unique document, allowing for direct lookup. ACID (our method) uses GPT-3.5 to create a sequence of abstractive keyphrases to serve as the document ID.

State-of-the-art generative retrieval models rely on document clustering to create document IDs, following the work of both Wang et al. (2022) and Tay et al. (2022), and the resulting document ID is an integer sequence corresponding to the clusters that the document belongs to. However, generating arbitrary sequences of integers is very different from what LMs are designed to do, since LMs are pretrained to generate natural language. In addition to negatively impacting LM generation performance, cluster-based integer IDs are not human-readable and require re-clustering if a substantial number of new documents are added to the index.

To address the issues with cluster-based IDs, we propose **ACID**, an abstractive, content-based ID assignment method for documents. Our method uses a language model (GPT-3.5 in our experiments) to generate a short sequence of *abstractive keyphrases* from the document's contents to serve as the document ID, rather than a hierarchical clustering ID or an arbitrary integer sequence. We find that ACID outperforms the strongest cluster-based IDs which are currently state-of-the-art for generative retrieval, and we also compare ACID to other methods for creating content-based IDs, such as taking the first 30 tokens of each document as its ID or choosing the top-30 keywords with respect to BM25 scores. We find that ACID generally outperforms cluster-based IDs and the other content-based IDs that we examined.

Finally, we show that adjusting decoding hyperparameters like beam width can improve retrieval performance with ACID, whereas cluster-based IDs only experience a marginal benefit with wider beams.

We provide our code for reproducing our results at `redacted-during-review`. In addition, we provide the precomputed IDs for all of the documents in the retrieval benchmark datasets in our experiments.

## 2 GENERATIVE RETRIEVAL WITH ACID

Since generative retrieval is a comparatively new approach for document retrieval, there is significant variation in the literature on how language models are trained to map queries to document IDs. Tay et al. (2022) distinguish between the 'indexing' step (where the LM is trained to link spans from the training, development, and test documents to their document IDs) and the 'finetuning' step (where the training query-document pairs are used to finetune the LM for retrieval). Note that generative retrieval models must index all documents, including the development and test documents, in order for the language model to be aware of their document IDs at inference time. Additionally, Wang et al. (2022) and Zhuang et al. (2023) perform data augmentation in the indexing and finetuning steps by introducing 'synthetic' queries, where a query generation model (Nogueira et al., 2019) based on T5 (Raffel et al., 2020b) generates additional queries for each document.

In the three subsections that follow, we elaborate on each of the steps for generative retrieval. Figure 2 depicts the steps needed to create our content-based document IDs, perform data augmentation, index the documents with the LM, and finetune the LM for generative retrieval.

Table 1: An example of a document, its cluster-based ID (where each level of the clustering has 10 clusters), and its associated natural language, content-based IDs. 'First $k$ tokens' sets the ID to be the document's first $k$ tokens. BM25 scoring uses the top-$k$ highest-scoring tokens from the document as the ID, where scores are based on Okapi BM25. ACID (our method) uses an LM (e.g., GPT-3.5) to generate 5 keyphrases, and the ID is the concantenation of those phrases.

**Document Text**

List of engineering branches Engineering is the discipline and profession that applies scientific theories , mathematical methods , and empirical evidence to design , create , and analyze technological solutions cognizant of safety , human factors , physical laws , regulations , practicality , and cost . In the contemporary era , engineering is generally considered to consist of the major primary branches of chemical engineering , civil engineering , electrical engineering , and mechanical engineering . There are numerous other engineering subdisciplines and interdisciplinary subjects that are derived from concentrations , combinations or extensions of the major engineering branches...

**Cluster-based Document ID**

9, 5, 1, 9, 6, 1, 0, 4, 8, 1, 3, 1, 2, 9, 0

**Content-based Document IDs**

| *First k Tokens* | *BM25 Scoring* | *ACID* |
|---|---|---|
| List of engineering branches Engineering is the discipline and profession that applies scientific theories , mathematical... | teletraffic optomechanical nano-engineering subdiscipline eegs biotechnical bioprocess mechatronics metallics crazing... | (1) Major engineering branches: chemical, civil, electrical, mechanical (2) Chemical engineering: conversion of raw materials with varied specialties (3) Civil engineering: design... |

## 2.1 DOCUMENT ID CREATION

In Table 1, we provide an example of a document about engineering sub-disciplines and the cluster-based and content-based IDs that would be derived from it. From the example, it is clear why we would expect ACID to outperform cluster IDs, since it is straightforward for LMs to generate the keyphrase sequence given an engineering-related query. The cluster ID, on the other hand, resembles an integer hash of the document (with some semantic information carried over from the clustering).

**Abstractive, Content-based IDs.** We create natural language IDs for every document to be indexed by generating keyphrases. Tokens from the document (up to the maximum context size of 4000 tokens) are used as part of a prompt to an LM to generate 5 keyphrases. The keyphrases are a brief abstractive summary of the topics in the document. The keyphrases are concatenated together to form the ACID for each document. We create IDs for every document in the training, development, and test sets.

We chose the GPT-3.5 API provided by OpenAI to generate keyphrases, though any reasonable pretrained LM can be used instead. The prompt that we used was:

> Generate no more than 5 key phrases describing the topics in this document.
> Do not include things like the Wikipedia terms and conditions, licenses, or
> references section in the list: (document body here)

**Cluster-based IDs.** By way of comparison with ACID, cluster-based IDs are integer sequences. An encoder creates an embedding vector for each document in the dataset, and the document embeddings are clustered using the $k$-means algorithm. If the number of documents in a cluster exceeds a predefined maximum, then subclusters are created recursively, until all subclusters contain fewer documents than the maximum. Each document's ID is a sequence of integers, corresponding to the path to the document through the tree of hierarchical clusters. The number of clusters at each level and the maximum number of documents in each cluster are hyperparameters. (For example, the values reported in Wang et al. (2022) were 10 and 100 respectively, which we also use in our experiments.)

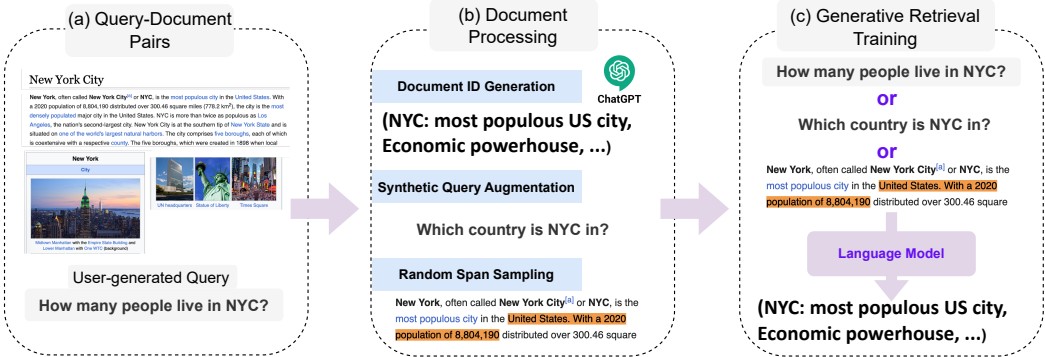

Figure 2: Data processing and model training. (a) Each document-query pair from the training corpus will be converted into inputs and outputs for finetuning the pretrained transformer decoder, which serves as the generative retrieval model. (b) GPT-3.5 is used to generate a sequence of keyphrases, which is used as the document ID. (c) Given a user query or a synthetic query, the generative retrieval model learns to generate the ID of the relevant document. We use a doc2query model to generate synthetic queries as additional inputs. Randomly sampled spans of 64 tokens can also be used as inputs to ensure that the model associates the contents of each document with its ID.

## 2.2 DOCUMENT INDEXING AND SUPERVISED FINETUNING

We first index all of the documents in the training, development and test sets. For indexing purposes, we consider two types of input/output pairs:

- (randomly selected document span, document ID)
- (synthetic query, document ID)

In other words, the LM is trained to generate the relevant document ID, given a randomly selected document span or a synthetic query, as part of the indexing task. We use a T5-based query generation model to provide synthetic queries given the body of each document. Note that, in our experiments, only synthetic queries are used during the indexing step. Although random document spans are used in other generative retrieval papers, we did not observe an improvement by doing so.

After document indexing, we finetune the model on the retrieval training data:

- (user-generated query, document ID)

In other words, the LM is trained to generate the document ID, given a real, user-generated query.

## 2.3 RETRIEVING DOCUMENTS

At inference time, the LM generates a document ID via beam search, given a user-generated query from the test set. We use a constrained decoder at inference time, which is constrained by a prefix tree such that it can only generate document IDs that exist in the corpus. Since each document ID maps to a unique document, it is straightforward to compute the proportion of queries for which the model retrieved the correct document. Model performance is measured based on the proportion of relevant documents retrieved within the top-1, top-10, and top-20 results in our experiments.

## 3 EXPERIMENTS

In the experiments below, we demonstrate that ACID outperforms cluster-based IDs on the NQ and MSMARCO retrieval benchmarks. Simpler forms of content-based IDs, like using the first 30 tokens of the document or BM25-based keyword selection, also outperform the cluster-based approach in most cases. We then show that ACID's outperformance persists across a range of language model sizes (as measured by the total number of parameters). Finally, we show that widening the beam

improves retrieval performance meaningfully for ACID, whereas cluster-based IDs benefit from beam width to a lesser degree (or not at all, in the case of the widest beam widths).

The BM25-based IDs were created by ranking all of the unique terms in each document by their BM25 scores, and taking the top 30 terms as the document ID. We used Anserini (Yang et al., 2017) to compute BM25 scores for the documents in each corpus. To avoid selecting very rare terms as part of each document's BM25-based document ID, we required that each term either appear at least 2 times in the document itself, or appear at least 5 times in the corpus.

We use the Natural Questions (NQ) (Kwiatkowski et al., 2019) and MSMARCO Bajaj et al. (2016) datasets. For each dataset, we finetune a pretrained language model for retrieval on 1k, 10k, and 100k random samples of the training split. Note that MSMARCO and NQ do not disclose their test sets publicly, and our results are reported on the provided development sets. Since we did not use the entirety of the training data that was available for NQ and MSMARCO, we created separate development sets for them by taking a random sample of each dataset's training data. We provide the details of each corpus in Table 2. Document length is highly variable, and we truncate all documents after 4k tokens.

We use the Pythia LMs (Biderman et al., 2023) to initialize the retrieval model in our experiments. All of our models are trained on AWS g5 instances equipped with Nvidia A10G GPUs. Models are optimized using AdamW (Loshchilov & Hutter, 2017). We provide the model hyperparameters used in the Appendix. The beam width for all experiments is 20, unless stated otherwise.

In Table 2, we provide the basic statistics for the NQ and MSMARCO datasets that we used. We deduplicate documents based on the first 512 tokens of each document, and documents with $\geq 95\%$ token overlap are considered duplicates.

Note that there is a substantial difference in the average document length between NQ and MS-MARCO datasets. While NQ and MSMARCO have queries of similar lengths, their document lengths are very different, since NQ documents are complete Wikipedia articles while MSMARCO passages are a few sentences long, excerpted from a longer document.

Table 2: Dataset characteristics. '# Pairs' refers to the number of query-document pairs. Average lengths refer to the average length in characters.

|  | # Pairs | Avg Query Length | Avg Doc Length |
|---|---|---|---|
| NQ-100k | 100,000 | | |
| NQ-Dev | 1,968 | 49.2 | 36,379.4 |
| NQ-Test | 7,830 | | |
| MSMARCO-100k | 100,000 | | |
| MSMARCO-Dev | 2,000 | 32.8 | 334.4 |
| MSMARCO-Test | 6,980 | | |

## 4 RESULTS

Note that there is substantial variation in the reported results on the NQ dataset among papers that use cluster IDs for generative retrieval. In Tay et al. (2022) and Wang et al. (2022), the top-1 accuracies with the NQ 320k dataset were 27.4% and 65.86% respectively, despite both groups using the same T5-Base model initialization and cluster-based ID approach. There are many possible explanations for the discrepancy (e.g., use of synthetic queries, computational budget, etc.), but at the time of writing, neither paper has made the code or processed data publicly available, which makes replication difficult. For this reason, we focus on internal comparisons rather than external ones, where we control the relevant experimental settings to ensure that the differences in results are meaningful.

### 4.1 NATURAL QUESTIONS

We evaluate our IDs on the NQ corpus in Table 3, and all results are based on a 160M-parameter Pythia LM. Across the NQ 1k, 10k, and 100k tasks, ACIDs outperform cluster-based integer IDs.

Table 3: Retrieval accuracy for Natural Questions. Accuracy refers to the percentage of queries in the evaluation set for which the ground-truth document ID was produced in the top-1, top-10, and top-20 candidates from constrained beam search decoding. NQ 1k, 10k, and 100k refer to the number of training query-document pairs used to finetune the LM.

|  | NQ 1k | | | NQ 10k | | | NQ 100k | | |
| --- | --- | --- | --- | --- | --- | --- | --- | --- | --- |
|  | Acc@1 | @10 | @20 | Acc@1 | @10 | @20 | Acc@1 | @10 | @20 |
| *Baselines* | | | | | | | | | |
| Cluster Integer IDs | 38.4 | 64.2 | 69.4 | 40.2 | 67.5 | 72.7 | 40.8 | 68.2 | 73.0 |
| First 30 Tokens | **41.9** | 66.0 | 69.9 | **43.3** | 67.6 | 71.6 | **47.7** | 71.2 | 74.4 |
| *Content-based IDs* | | | | | | | | | |
| BM25 Top-30 | 36.5 | 66.1 | 70.9 | 36.8 | 66.1 | 71.1 | 37.0 | 68.2 | 72.8 |
| ACID | 39.2 | **69.2** | **74.0** | 40.5 | **70.7** | **75.2** | 40.9 | **71.2** | **75.9** |

The simple approach of using the first 30 tokens from each document to create IDs also outperforms the cluster-based approach.

We observed that the top-1 accuracy with the first 30 tokens as the ID is quite high. This may be due to the structure of the NQ documents, which are Wikipedia articles. The first tokens of every document are the title of the Wikipedia page, and so the first 30 tokens represents a very effective ID for retrieval purposes. Nonetheless, ACIDs outperform the first 30 tokens at the top-10 and top-20 accuracies.

## 4.2 MSMARCO

Table 4: Retrieval accuracy for MSMARCO. Accuracy refers to the percentage of queries in the evaluation set for which the ground-truth document ID was produced in the top-1, top-10, and top-20 candidates from constrained beam search decoding. MSMARCO 1k, 10k, and 100k refer to the number of training query-document pairs used to finetune the LM.

|  | MSMARCO 1k | | | MSMARCO 10k | | | MSMARCO 100k | | |
| --- | --- | --- | --- | --- | --- | --- | --- | --- | --- |
|  | Acc@1 | @10 | @20 | Acc@1 | @10 | @20 | Acc@1 | @10 | @20 |
| *Baselines* | | | | | | | | | |
| Cluster Integer IDs | 41.1 | 59.5 | 64.2 | 42.4 | 62.3 | 67.1 | 46.8 | 68.8 | 73.4 |
| First 30 Tokens | 49.0 | 73.0 | 77.8 | 48.7 | 72.8 | 77.9 | 51.8 | 76.0 | 79.6 |
| *Content-based IDs* | | | | | | | | | |
| BM25 Top-30 | 48.7 | 74.3 | 79.4 | 49.1 | 75.7 | 80.1 | 52.0 | 79.2 | 82.9 |
| ACIDs | **49.1** | **74.3** | **80.1** | **50.4** | **76.3** | **80.4** | **52.9** | **79.5** | **84.0** |

Our experiments on the MSMARCO task confirm the conclusions that we drew from the results on the NQ task. We present the results in Table 4, and all results are based on a 160M pretrained Pythia LM. The ACIDs offer better retrieval performance compared to the other ID generation techniques.

## 4.3 MODEL SIZE

We examine whether the relative outperformance of ACIDs versus cluster IDs is affected by the number of parameters in the model. Our default experiments in the previous sections used 160 million-parameter Pythia models, and in Figure 3 we conduct experiments going up to 2.8 billion-parameter models.

We observe that ACIDs continue to outperform cluster IDs, even as we vary the model size. In general, increasing the size of the model leads to an improvement in retrieval performance, regardless of the ID type.

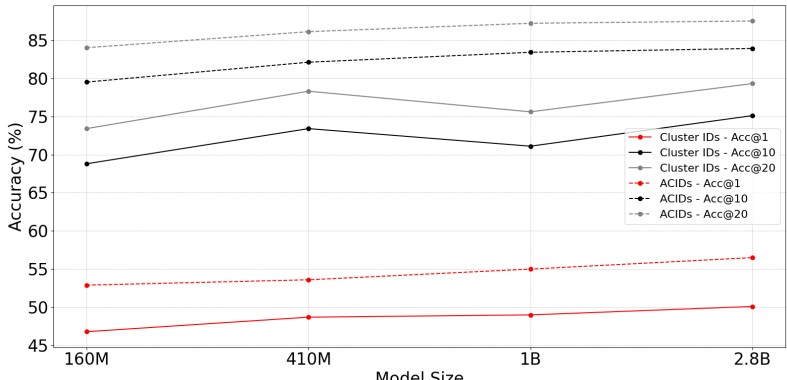

Figure 3: Retrieval accuracy versus the number of model parameters on the MSMARCO 100k dataset.

## 4.4 BEAM WIDTH

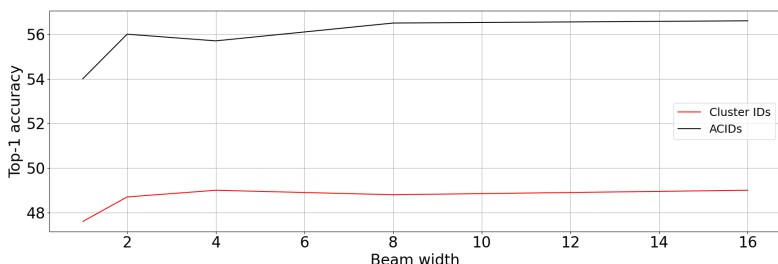

Figure 4: Top-1 accuracy of the 1B-parameter model versus beam width on the MSMARCO 100k dataset.

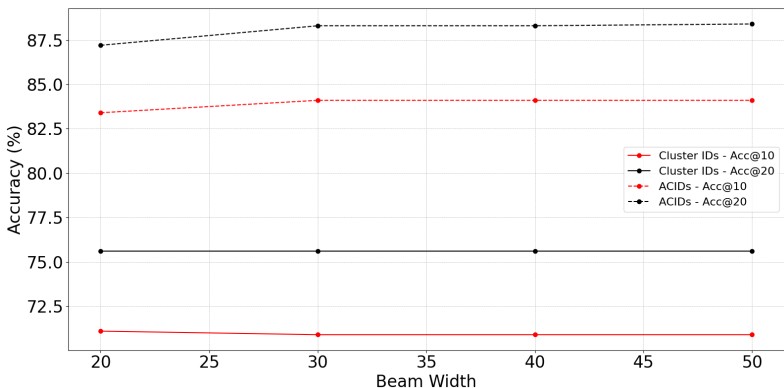

Figure 5: Top-10 and 20 accuracies of the 1B-parameter model versus beam width on the MS-MARCO 100k dataset.

From Figure 4, we see that larger beam widths generally improves retrieval accuracy on MS-MARCO, though with rapidly diminishing returns. The top-1 accuracy does not benefit past a beam width of 8, and the accuracy rapidly plateaus as beam width increases from 1 to 16. This is true for both cluster IDs and ACID, though ACID does benefit more in absolute terms than cluster IDs from a wider beam (when comparing a beam width of 1 to a beam width of 16).

In Figure 5, we examine the effect of very wide beams on accuracy at 10 and 20 for the MSMARCO dataset. Some benefit is observed when ACID is the document ID, but no improvement is observed for cluster IDs.

As discussed previously, the cluster ID is typically restricted to a small number of clusters per level (the digits 0 through 9, for example), and so a wide beam in excess of that number doesn't yield any improvements, whereas ACID does benefit from wider beams, since it is a natural-language ID with access to the full vocabulary of the LM.

## 5 RELATED WORK

Tay et al. (2022) explore a number of techniques for creating document IDs for generative retrieval, including atomic document IDs, randomly assigned integer IDs, and semantic IDs based on hierarchical clustering. The last technique was found to be the most effective, where the document IDs with were formed via hierarchical $k$-means clustering on BERT-based document vectors. The main difference between that approach and ours is that, during finetuning, their approach requires learning the "semantics" of the cluster IDs, while ours uses natural language phrases that are already in some sense familiar to the pretrained model. Wang et al. (2022) also used IDs based on hierarchical clustering with BERT embeddings and proposed the prefix-aware weight-adaptor (PAWA) modification, where a separate decoder was trained to produce level-specific linear projections to modify the ID decoder's outputs at each timestep. The authors also incorporated synthetic queries from a doc2query model to augment the user-generated queries in the dataset. Pradeep et al. (2023) scale the cluster ID-based approach to generative retrieval to millions of documents, and explore the impact of adding synthetic queries for documents that do not have a query sourced from a user.

The aforementioned papers used IDs that were not optimized for the retrieval task, but other work has explored creating document IDs in a retrieval-aware manner. In Sun et al. (2023), the document IDs are treated as a sequence of fixed-length latent discrete variables which are learned via a document reconstruction loss and the generative retrieval loss. However, this method does experience collisions as some documents are assigned to the same latent integer ID sequence, though the authors do not report the collision rate in the paper.

Bevilacqua et al. (2022) proposed a model that, given a query, generates the n-grams that should appear in the relevant documents. All documents that contain the generated n-grams are then retrieved and reranked to produce the final search results. (This is in contrast our approach, which seeks to associate a unique ID to each document for generative retrieval.) The authors propose several methods for reranking based on n-gram scores produced by the LM. However, the n-gram generation and reranking approach does not always outperform the dense retrieval baseline.

In addition, there is a substantial body of work that involves model-generated text and retrieval. De Cao et al. (2020) generate the text representation of entities autoregressively instead of treating entities as atomic labels in a (potentially very large) vocabulary. Nogueira et al. (2019) use an encoder-decoder model to generate synthetic queries for each document in the index and concatenate them together to improve retrieval performance. The expanded documents are indexed using Anserini and BM25. Synthetic queries from these 'doc2query' models are also used for data augmentation in generative retrieval. Mao et al. (2020) use pretrained language models to expand queries with relevant contexts (e.g., appending the title of a relevant passage to the query, etc.) for retrieval and open-domain question answering.

## 6 CONCLUSION

We have shown that abstractive keyphrases are excellent document IDs for generative retrieval. Our results show a clear improvement in retrieval performance on the Natural Questions and MSMARCO datasets versus the state-of-the-art cluster-based integer IDs. To the best of our knowledge, ACID is the first example of a human-readable document ID that consistently outperforms cluster IDs on standard IR benchmarks. The choice of ID for generative retrieval is clearly a major factor in model performance, and we expect that future work will explore other possibilities for creating effective natural-language document IDs.

We also hypothesize that natural-language IDs will scale better than cluster integer IDs as the number of documents that need to be indexed increases. To the best of our knowledge, the largest-scale study on generative retrieval (Pradeep et al., 2023) examined retrieval performance on all 8.8M documents in MSMARCO, but the documents in that dataset are very short. On an uncompressed basis, MSMARCO uses 4.3GB of storage space, whereas a larger web-scale dataset like C4 (Raffel et al., 2020a) uses 750GB by comparison. Memorizing sequences of words should be a more efficient use of model capacity than memorizing sequences of integers, given how LMs are pretrained. As such, we expect that as the size of the retrieval corpus expands to something closer to web-scale, the ID that LMs use for representing documents will have a significant effect on retrieval performance.

ACKNOWLEDGMENTS

Redacted during review.

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

# A    APPENDIX

## A.1    MODEL HYPERPARAMETERS

We used the Pythia (Biderman et al., 2023) series of pretrained LMs for our generative retrieval experiments. The hyperparameters are provided in the table below.

Table 5: Model hyperparameters.

| Parameters | {160M, 410M, 1B, 2.8B} |
|---|---|
| Learning Rate | 1e-4 |
| Layers | {12, 24, 16, 32} |
| Pretraining Corpus | The Pile (Gao et al., 2020) |
| Model Dim | {768, 1024, 2048, 2560} |
| Attn Heads | {12, 16, 8, 32} |
| Optimizer | AdamW |

