# OpenReview forum: "ACID: Abstractive, Content-Based IDs for Document Retrieval with Language Models"
_ICLR.cc/2024/Conference — ICLR 2024 Conference Withdrawn Submission_

### Official Review · Reviewer_mfex · 2023-10-28

**Soundness:** 2 fair
**Presentation:** 2 fair
**Contribution:** 2 fair
**Rating:** 3
**Confidence:** 5

**Summary:**

The paper studies the problem of document semantic IDs for generative retrieval. The authors propose a method called abstractive, content-based IDs, where they prompt LLM to write keywords to summarize the documents and serve them as identifiers for generative retrieval. Experiments are conducted on two datasets to verify the effectiveness of their proposed method.

**Strengths:**

1. The research problem is very important and interesting.
2. The proposed method is sound.

**Weaknesses:**

1. The paper is not complete. Although it is not a hard requirement that an ICLR submission should have 9 full pages, people usually serve it as a common sense of measure of the completeness of the submission.
2. Absence of comparison with several baselines.
3. Some details are not provided, e.g., the size of the corpus in each dataset.
4. Generalization to later real-world datasets.

**Questions:**

1. How many documents are there in your NQ and MSMACRO datasets? It seems to me that you need to access GPT3.5 to generate semantic IDs for each document, which will be time-consuming and expensive for large datasets in real-world (with millions of documents in the corpus).
2. The idea of using words or keywords as identifiers for documents is not new. There are some existing works with a similar philosophy [1,2,3,4]. Can I ask why you don’t compare with them?
3. What is the average length of your generated IDs for documents?


[1] Autoregressive Search Engines: Generating Substrings as Document Identifiers. NeurIPs 2022.

[2] Term-Sets Can Be Strong Document Identifiers For Auto-Regressive Search Engines. Arxiv 2023.

[3] A Unified Generative Retriever for Knowledge-Intensive Language Tasks via Prompt Learning. SIGIR 2023.

[4] Semantic-Enhanced Differentiable Search Index Inspired by Learning Strategies. KDD 2023.

---

### Official Review · Reviewer_qncb · 2023-10-30

**Soundness:** 2 fair
**Presentation:** 3 good
**Contribution:** 2 fair
**Rating:** 5
**Confidence:** 4

**Summary:**

Generative retrieval models learn a sequence-to-sequence model to generate the document's ID given the input query. Previous work typically construct the document ID as a sequence of semantic codes obtained from hierarchical clustering. This paper introduces ACID, which represents each document's ID as generated keywords from a large language model.  Empirically, on the MS-MARCO and NQ datasets with self-constructed train/test split, ACID significantly improves top-10 accuracy versus the baselines.

**Strengths:**

1. Writing of this paper is easy to follow

**Weaknesses:**

1. Limited novelty, as the learning method is quite similar to DSI (Tay et al 2022) and NCI (Wang et al 2022).

2. Lack of comparing baselines, see follow-up questions below.

3. Lack of ablation studies for the effect of data augmentation see follow-up questions below.

**Questions:**

### 1. Incremental Document-up Setup
The novelty of this work can be enhanced if there's empirical study on how ACID perform on the incremental document update setup, where DSI and NCI are not capable of (require re-clustering, as noticed by the author in first paragraph on Page 2). How does ACID perform on new document IDs, without re-training the models?


### 2. Lack of comparing baselines
(1) Why not following the same train/test split as NCI, as the author of NCI released their dataset/code since Nov, 2022 [https://github.com/solidsea98/Neural-Corpus-Indexer-NCI]? For example, they have pre-processed datasets available at least for NQ and TriviaQA.

(2) The author should compare ACID with other retrieval baselines, such as the embedding-based retrieval methods (e.g., DPR/ANCE) and sparse retrieval methods (e.g., BM25/BM25+DocT5Query), as reported in Table 1 and Table 2 of NCI paper [Wang et al, 2022)?

### 3. Lack of ablation studies for the effect of data augmentation
It is not clear how much performance gain is from the generated keyphrases or the amount of data augmentations. Specifically,

(1) The author should consider an ablation experiment to see how the model performs versus the amount of data augmentation used. The data augmentation refers to the amount of (synthetic query, document ID) pairs.

(2) If the proposed method obtain performance gain from using data augmentations, then the comparing baselines should also use the same amount of data augmentation. Otherwise, the comparison is not very fair.

---

### Official Review · Reviewer_2EQH · 2023-10-31

**Soundness:** 3 good
**Presentation:** 3 good
**Contribution:** 2 fair
**Rating:** 5
**Confidence:** 2

**Summary:**

Generative retrieval is an interesting research topic that changed the way researchers used to conduct information retrieval. The key idea is to digress from traditional cosine-similarity-based computation where documents and information needs were separately encoded into a more homogeneous encoding. While there were issues in the initial models as reported by this paper, this paper further improves the state-of-the-art by automatically generating abstract keywords associated with a document using a language model.

The abstract keywords seem interesting and from the prompt, the authors tend to automatically extract "topics" for every document. Given that there have been works surrounding topics and transformers, the abstraction keywords could very well be latent topics, e.g., Thompson L, Mimno D. Topic modeling with contextualized word representation clusters. arXiv preprint arXiv:2010.12626. 2020 Oct 23. and there are some follow-up published works by other authors.

What happens in this work is that the authors gather a certain number of keywords using a language model. Then these keywords are used as IDs replacing the previous efforts.

**Strengths:**

The key strength of the paper is modelling topics in documents. Topics have demonstrated strong retrieval performance even under traditional information retrieval settings.

The paper demonstrates that it improves upon existing strong comparative models.

**Weaknesses:**

It would have been excellent had the authors compared their methods by extracting topics for every document using a traditional topic model such as the latent Dirichlet allocation or its variants where phrases are automatically extracted. The key advantage is that one does not need to perform computationally expensive prompting tasks as executed by the authors in section 2.1

One of the key challenges is how many keyphrases to select to encode for retrieval. The paper will further improve if the authors compare and conduct experiments by choosing a varying number of keywords in the document.

**Questions:**

While I do not have any specific questions in this section, I would request the authors to help me understand how the key weaknesses in the paper could be addressed.

---

### Official Review · Reviewer_xBCe · 2023-11-03

**Soundness:** 2 fair
**Presentation:** 3 good
**Contribution:** 2 fair
**Rating:** 5
**Confidence:** 3

**Summary:**

In this paper the authors propose a new method for generative retrieval. Instead of using semantic cluster IDs as document ID in previous works (e.x. DSI, NCI) the new method ACID utilizes GPT 3.5 to generate 5 key phrases as document ID. Empirical study compares cluster integer IDs with different types of token based IDs and the proposed method achieves better Acc@10 and @20 on NQ-100K and MSMARCO datasets.

**Strengths:**

* The paper presents a novel method for document indexing ACID that outperforms cluster based IDs on generative retrieval tasks.
* The presentation of the paper is easy to follow and discussion of related works seems extensive.

**Weaknesses:**

* **Regarding the experiments**.
  * The paper claims the ACID achieves better performance than the counterparts DSI, NCI. Yet there's no apple to apple comparison between these methods. The only empirical results are "internal comparisons". From these results it's hard to directly draw the conclusions that ACID is better than DSI/NCI as there could be many discrepancies between implementation details.
  * For retrieval tasks it's common to report Recall@k whereas the only metric reported is Acc@k. It's not clear how the proposed method actually performs compared with other retrieval baselines.

* **Regarding the method**.
  * The cluster index based document ID would ensure there are no duplication in document IDs, as each document is an unique leaf in the hierarchical clustering tree. This is not the case for ACID. As the indexing space grows, there could be more and more chances that multiple similar documents getting the same document ID.
  * In clustering based document ID setting, similar documents would share document id prefix since they are closer in the hierarchical clustering tree. However, in ACID this is also not the case. I suspect this is why the beam size is much more important in ACID because retrieved documents are more scattered than clustering based methods.

**Questions:**

* Why 5 phrases for each document?
* How do ACID ensure each document ID is unique?
* It seems the NQ and MSMARCO document space has ben altered to avoid similar documents. Why?
* How does the ACID perform in terms of recall?